# Characteristics of SARS-CoV-2 Infection in an Actively Monitored Cohort of Patients with Lupus Nephritis

**DOI:** 10.3390/biomedicines10102423

**Published:** 2022-09-28

**Authors:** Bogdan Obrișcă, Alexandra Vornicu, Roxana Jurubiță, Valentin Mocanu, George Dimofte, Andreea Andronesi, Bogdan Sorohan, Camelia Achim, Georgia Micu, Raluca Bobeică, Constantin Dina, Gener Ismail

**Affiliations:** 1Department of Nephrology, Fundeni Clinical Institute, 022328 Bucharest, Romania; 2Department of Nephrology, “Carol Davila” University of Medicine and Pharmacy, 020021 Bucharest, Romania; 3Department of Anatomy, Ovidius University, 900470 Constanța, Romania

**Keywords:** lupus nephritis, SARS-CoV-2 infection, corticosteroids, hydroxychloroquine

## Abstract

(1) Background: We sought to investigate the impact of the COVID-19 pandemic in patients with lupus nephritis (LN); (2) Methods: A total of 95 patients with LN actively monitored in our department between 26 February 2020, when the first case of COVID-19 was diagnosed in Romania, and 1 May 2021, were included in the study. Multivariate logistic regression analysis was performed to identify the independent risk factors for SARS-CoV-2 infection; (3) Results: A total of 15 patients (15.8%) had a confirmed SARS-CoV-2 infection during a total follow-up time of 105.9 patient-years (unadjusted incidence rate: 14.28 SARS-CoV-2 infections per 100 patient-years). Median time to SARS-CoV-2 infection was 9.3 months (IQR: 7.2–11.3). The majority of patients had a mild form of SARS-CoV-2 infection (73.3%), while the remaining had moderate forms. None of the patients had a severe infection or a SARS-CoV-2-related death. The most frequent symptom was fatigue (73.3%), followed by loss of taste/smell (53.3%) and fever (46.7%). Forty percent of those with SARS-CoV-2 infection were hospitalized for a median 11.5 days (IQR:3.75–14). In the multivariate logistic regression analysis, a current oral corticosteroid dose ≥ 15 mg/day was associated with a 7.69-fold higher risk (OR, 7.69; 95%, 1.3–45.46), while the use of hydroxychloroquine was associated with a 91% lower risk for a SARS-CoV-2 infection (OR, 0.09; 95%CI, 0.01–0.59). (4) Conclusions: Our study confirms that the SARS-CoV-2 infection-associated morbidity might only be moderately increased in patients with LN. The current oral corticosteroid dose was the only independent predictor of infection occurrence, while use of hydroxychloroquine was associated with a protective effect.

## 1. Introduction

Since the emergence of SARS-CoV-2 (severe acute respiratory syndrome coronavirus 2) infection at the end of 2019, over 580 million confirmed cases and over 6.4 million deaths have been reported worldwide [1]. Certain populations, including elderly patients and those with chronic conditions (e.g., chronic kidney disease), appear to be at higher risk for severe COVID-19 [2]. 

Lupus nephritis (LN) affects nearly 60% of patients with systemic lupus erythematosus (SLE) and is associated with substantial morbidity and mortality [3,4]. Patients with LN are known to be at higher risk for severe infections due to both an underlying immune dysfunction and as a consequence of immunosuppressive therapy (IS) [5]. We have previously shown that there is a significant treatment-related morbidity in LN, as these patients have an incidence rate of 26.6 infections and 9.56 severe infections per 100 patient-years [6]. Treatment with high-dose oral corticosteroids was identified as the most important contributor being associated with an up to 7-fold higher risk for infection occurrence [6]. Accordingly, there is an increased concern that patients with SLE might be at higher risk for a severe SARS-CoV-2 infection [7]. Several observational cohorts have outlined that the COVID-19-related morbidity might only be moderately increased in patients with SLE [8,9,10]. Nonetheless, a nationwide epidemiologic study suggested that, despite the overall mortality of patients with SLE and SARS-CoV-2 infection was lower compared to the total population admitted for SARS-CoV-2 infection, the mortality rate in younger patients was significantly increased [7]. However, many studies reported patients with inactive disease and collected data only by telephone interviews or online questionnaires, which may limit the interpretation of these results [7,8,9]. Additionally, data regarding the impact of SARS-CoV-2 infection among patients with LN are scarce [11]. There is only one report that included 101 patients with LN, of whom two developed a mild infection, from Wuhan, at the beginning of the pandemic [11].

We sought to investigate the characteristics of SARS-CoV-2 infection and the associated risk factors in an actively monitored cohort of patients with LN throughout an observation period spanned over the first three waves of COVID-19 pandemic.

## 2. Materials and Methods

### 2.1. Study Population and Data Collection

This is a prospective, observational study that enrolled all patients with lupus nephritis actively monitored between 26 February 2020 (when the first case of SARS-CoV-2 infection was diagnosed in Romania) and 1 May 2021, in the Nephrology Department of Fundeni Clinical Institute, Bucharest, Romania. The patients had to fulfill the classification criteria of either the 1997 American College of Rheumatology Classification [12], the 2012 Systemic Lupus International Collaborating Clinics Classification [13] or the 2019 European League Against Rheumatism/American College of Rheumatology Classification [14], according to their period of SLE diagnosis. Only patients with signs of renal involvement (proteinuria greater than 500 mg/day, active urinary sediment, increased serum creatinine) or biopsy proven LN were included in the analysis. Patients without any clinical or laboratory assessments during the study period or lost to follow-up were excluded from the analysis.

The clinical variables were obtained by reviewing the patients’ medical records and included demographic data, the history of SLE organ involvement, as well as details of past and current immunosuppressive regimens. Laboratory data included serum creatinine, the estimated glomerular filtration rate (eGFR, calculated by CKD-EPI equation), serum albumin (g/dL), 24-h proteinuria (g/day), hematuria (cells/mmc), hemoglobin (g/dL), serum fibrinogen (mg/dL), C-reactive protein (mg/L), serum complement levels (C3 and C4, mg/dL), complete cell blood count and serum immunoglobulin levels (IgA, IgG and IgM; mg/dL). The baseline variables were considered those that were within 3 months (before or after) of the pandemic onset. 

The study was conducted with the provisions of the Declaration of Helsinki and the protocol was approved by the local ethics committee (The Ethics Council of Fundeni Clinical Institute, Registration number: 20639; date of approval: 26 February 2020). All patients provided informed consent before study entry.

### 2.2. SARS-CoV-2 Infection Diagnosis and Assessment of Severity

Patients were screened for SARS-CoV-2 infection via outpatient visits, hospital admissions, periodic telephone interviews and a thorough review of the electronic clinical health records during the study follow-up from 26 February 2020 to 1 May 2021. Screening for SARS-CoV-2 infection also included periodic testing by real-time reverse transcription polymerase chain reaction (RT-PCR) and/or antigen tests. A positive antigen test for SARS-CoV-2 was followed by a confirmatory RT-PCR.

A COVID-19 diagnosis was considered as confirmed if typical clinical symptoms were accompanied by either positive nasopharyngeal swabs for SARS-CoV-2, as determined via RT-PCR, or by a positive serological test. In the case of confirmed infections, data were collected regarding clinical course, need for hospitalization, treatment and requirement for oxygen therapy or ventilatory support. 

SARS-CoV-2 infection severity was assessed as defined by the COVID-19 Treatment Guidelines Panel of the National Institutes of Health as mild illness (individuals who have any of the various signs and symptoms of COVID-19 but who do not have shortness of breath, dyspnea, or abnormal chest imaging), moderate illness (individuals who show evidence of lower respiratory disease during clinical assessment or imaging and who have an oxygen saturation ≥94% on room air at sea level) and severe illness (individuals who have oxygen saturation <94% on room air at sea level, a ratio of arterial partial pressure of oxygen to fraction of inspired oxygen <300 mmHg, respiratory frequency >30 breaths/min or lung infiltrates >50%) [15].

### 2.3. Statistical Analysis

Continuous variables were expressed as either mean (±standard deviation) or median (IQR, interquartile range) and categorical variables as percentages. Differences between groups were assessed in case of continuous variables by Student *t* test or Mann–Whitney test, according to the distribution of the dependent variable and in the case of categorical variables by Pearson χ^2^ test. The probability of infection-free survival was assessed by the Kaplan–Meier method and the log-rank test was used for comparisons. Univariate and multivariate logistic regression analyses were performed to identify the independent predictors of SARS-CoV-2 infection. The results of logistic regression are expressed as an odds ratio (OR) and the corresponding 95% confidence interval (95% CI). Identification of the corticosteroid dose that would most accurately predict a SARS-CoV-2 infection was undertaken by receiver-operating curve [area under the curve (AUC), sensitivity, specificity, positive (LR+) and negative (LR−) likelihood-ratio]. In all analyses, the *p* values are two-tailed and all *p* values less than 0.05 were considered statistically significant.

Statistical analyses were performed using the SPSS program (SPSS version 20, Chicago, IL, USA) and MedCalc^®^ Statistical Software (version 20.011; MedCalc Software Ltd., Ostend, Belgium; https://www.medcalc.org; 2021).

## 3. Results

### 3.1. Study Population

Among the 110 patients with LN considered for study inclusion, 95 patients were actively monitored between 26 February 2020 (when the first case of SARS-CoV-2 infection was diagnosed in Romania) and 1 May 2021. Fifteen patients were excluded from the analysis because they were lost to follow-up (n = 7), did not have any clinical or laboratory assessments during the study period (n = 5) or died prior to the COVID-19 pandemic (n = 3). 

The study population had a mean age of 39 ± 14 years, 83.2% were females and had a duration of SLE prior to study enrollment of 7.3 years (IQR: 2.3–13.6). Articular (67.4%), cutaneous (61.1%) and serosal (22.1%) involvement were among the most frequent sites of extrarenal involvement encountered. At the time of the COVID-19 pandemic start, the study cohort had an eGFR of 69 ± 32 mL/min/1.73 m^2^ and a 24-h proteinuria of 0.3 g/day (IQR: 0–1.2).

In terms of treatment, 83.2% of patients were on an immunosuppressive regimen, of whom the majority (74.7%) were on maintenance therapy. Nonetheless, 8.4% of the study cohort consisted of patients newly diagnosed with LN and were currently during induction therapy. Finally, 63.2% of patients were in clinical remission.

### 3.2. SARS-CoV-2 Infection Characteristics

A total of 15 patients (15.8%) had a confirmed SARS-CoV-2 infection during a total follow-up time of 105.9 patient-years (Table 1). The unadjusted incidence rate was 14.28 SARS-CoV-2 infections per 100 patient-years. The median time to SARS-CoV-2 infection was 9.3 months (IQR: 7.2–11.3). 

The majority of patients had a mild form of SARS-CoV-2 infection (73.3%), while the remaining had moderate forms. None of the patients had a severe infection or a SARS-CoV-2-related death. There were two deaths throughout the study period in patients without any current or prior history of SARS-CoV-2 infection. The most frequent symptom was fatigue (73.3%), followed by loss of taste/smell (53.3%) and fever (46.7%) (Figure 1). Forty percent of those with SARS-CoV-2 infection were hospitalized for a median 11.5 days (IQR:3.75–14). None of the patients needed non-invasive or invasive ventilation. 

In terms of SARS-CoV-2 vaccination, at some point during the late follow-up period, 41.5% of patients received at least one dose of vaccine. However, the patients were more likely vaccinated if they had a previous SARS-CoV-2 infection compared to patients that did not develop an infection (66.7% vs. 36.7%, *p* = 0.04).

In terms of LN evolution post-SARS-CoV-2 infection, 20% patients had an increase in serum creatinine of at least 0.5 mg/dL (but less than 1 mg/dL), while 13.3% and 6.6% had an increase in the level of proteinuria and hematuria, respectively (Table 1). However, none needed an augmentation of IS therapy following COVID-19.

### 3.3. Risk Factors for SARS-CoV-2 Infection

The univariate analysis regarding the risk factors associated with SARS-CoV-2 infection is provided in Table 2. Patients with SARS-CoV-2 infection had a lower serum albumin level (3.5 ± 1.2 g/dL), a tendency for higher proteinuria (0.8 g/day; IQR: 0.4–2.5) and a higher C-reactive protein (14.3 ± 29.5 mg/L) compared to patients without SARS-CoV-2 infection (Figure 2). There were no statistically significant differences in terms of the severity of renal dysfunction or type of extrarenal involvement. 

When analyzing the impact of the immunosuppressive regimen, we identified that the current oral corticosteroid dose had the strongest association with the risk of SARS-CoV-2 infection, and those that developed the infection had a significantly higher dose (16 mg/day, IQR: 7–21) compared to those without infection (6 mg/day, IQR: 4–10) (Figure 2). Accordingly, we did a sensitivity analysis to identify the corticosteroid dose that would most accurately predict the occurrence of SARS-CoV-2 infection through a receiver-operating curve (AUC = 0.77; *p* = 0.003) (Figure 3). A current oral corticosteroid dose ≥ 15 mg/day had the best performance characteristics for the prediction of SARS-CoV-2 infection occurrence with a sensitivity of 60% (95%CI, 26.2–87.8%) and specificity of 91.6% (95%, 80–97.7%). Additionally, the LR+ and LR- were 7.2 (95%CI, 2.48–20.91) and 0.44 (95%, 0.2–0.94).

Furthermore, a significantly higher proportion of patients with a current oral corticosteroid dose ≥ 15 mg/day developed SARS-CoV-2 infection (Figure 4). The use of hydroxychloroquine conferred a protective effect against SARS-CoV-2 infection (Figure 4, Table 2). Additionally, those with SARS-CoV-2 infection were more likely to be during induction therapy and less likely to be in clinical remission compared to those that did not develop the infection (Figure 4). 

In multivariate logistic regression analysis, a current oral corticosteroid dose ≥ 15 mg/day was associated with a 7.69-fold higher risk for SARS-CoV-2 infection (OR, 7.69; 95%, 1.3–45.46) (Table 3). Additionally, the use of hydroxychloroquine was significantly associated with a 91% lower risk for SARS-CoV-2 infection (OR, 0.09; 95%CI, 0.01–0.59).

## 4. Discussion

Our study confirms that the SARS-CoV-2 infection-associated morbidity might only be moderately increased in patients with LN. We identified an incidence rate of 14.28 SARS-CoV-2 infections per 100 patient-years. The majority of infections were mild and there were no COVID-19-attributed deaths. A current oral corticosteroid dose of over 15 mg/day significantly increased the risk for SARS-CoV-2 infection, while the use of hydroxychloroquine might possess an infection protective effect. In addition, given that vaccination was an ongoing process and started in the late follow-up period, these results reflect the characteristics of SARS-CoV-2 infection in a mainly unvaccinated cohort.

Since the emergence of SARS-CoV-2 infection, a particular attention has been directed towards patients with autoimmune disorders, known to be at increased risk of severe infections due to both the underlying immune dysfunction and as a consequence of immunosuppressive therapy [5,16]. Among these, the relationship between SLE and SARS-CoV-2 infection is of particular interest as one of the central pathogenic pathways (type I interferon system) closely resembles the immune response to viral infections [4,17]. As such, enhanced signaling through type I interferon (IFN) system seen in SLE might contribute to COVID-19 pathogenesis through distinct mechanisms, either by a protective, antiviral effect, or by promoting the hyperinflammatory response seen with SARS-CoV-2 infection [17]. Several observational studies provided conflicting results regarding the outcome of patients with SLE and SARS-CoV-2 infection. Two reports, from Spain and Italy, have suggested that the SARS-CoV-2-related morbidity is only moderately increased in patients with SLE [8,9]. These results must be interpreted with caution, as the studies were conducted by telephone interviews or web-based surveys and might have included patients with inactive disease or mild forms of SARS-CoV-2 infection and missed the COVID-19-related deaths. On the contrary, a nationwide study conducted in France among hospitalized patients with SLE during COVID-19 pandemic (n = 11,055, of whom 1411 had confirmed COVID-19) suggested that, although the mortality rate was lower compared to the general population admitted for COVID-19 (9.5% vs. 15.7%, *p* < 0.0001), the mortality rate at a younger age tended to be higher in those with SLE [7]. Similarly, a report from the COVID-19 Global Rheumatology Alliance showed that the hospitalization rate for patients with SLE and SARS-CoV-2 infection was higher compared to other autoimmune disorders [16]. Chronic comorbidities (chronic kidney disease, chronic pulmonary disease and cardiovascular disease) and history of LN were among the predictors of poor outcome for COVID-19 in patients with SLE [7]. 

Given the severity of target organ damage and the intensity of immunosuppressive therapy, patients with SLE and LN might be at an increased risk of a poor outcome due to COVID-19 compared to those with SLE without renal involvement [6]. In terms of the overall infection risk, we have previously shown that patients with LN have an unadjusted incidence rate of 26.6 infections and 9.56 severe infections per 100 patient-years [6]. Among these, the pulmonary infections were the most frequently encountered with an unadjusted incidence rate of 10.7 events per 100 patient-years [6]. In this analysis, we identified an incidence rate of 14.28 SARS-CoV-2 infections per 100 patient-years. Given that the majority were mild forms of SARS-CoV-2 infection, this incidence is lower compared the overall infectious risk seen in the previous study, but similar to that of pulmonary infections. To our knowledge, only one previous study specifically addressed the impact of COVID-19 in patients with LN [11]. In this report, among the 101 patients with LN that responded to questionnaires or telephone interviews, there were two confirmed cases of COVID-19, both mild and without any need for supplemental oxygen [11]. In addition, several studies outlined that, in patients with SLE, the presence of LN is associated with a poor COVID-19 outcome [7,18]. It is worth mentioning that, compared with previous reports, this was a prospective study that included only patients that were actively monitored for SARS-CoV-2 infection by outpatient or inpatient visits, telephone interviews and periodic viral testing through RT-PCR or antigen tests. Moreover, while other cohorts generally reported patients with clinically inactive disease, in our study, 8.4% of patients were undergoing induction therapy and approximately 37% had signs of clinical activity, suggesting that this cohort is characterized by more activity and, therefore, might be at higher risk for SARS-CoV-2-infection. Additionally, the majority of studies undertook a surveillance in patients with autoimmune disorders (including SLE) for SARS-CoV-2 infection that occurred during a short period of observation (ranging 1–9 months) [5,7,8,9,10,18,19,20]. To our knowledge, this is the cohort of patients with SLE and LN with the longest follow-up period spanning the first three waves of the COVID-19 pandemic (mean follow-up 13.5 ± 2 months).

In terms of risk factors for COVID-19, the intensity of a current immunosuppressive regimen was the most important contributor to the infection burden as those that were during induction therapy or were in clinical remission had a higher (by approximately 4-fold) or, respectively, lower (by approximately 70%) risk of SARS-CoV-2 infection. Regarding the immunosuppressive or immunomodulatory agents, we identified two directions. First of all, the current oral corticosteroid dose was strongly associated with infection risk. This is in line with our previous study that showed an increased risk in LN for any infection and severe infections by 5.1-fold and 7.5-fold, respectively, with an induction therapy that included an initial high-dose oral steroid regimen (≥0.5 mg/kg/day) [6]. In a recent meta-analysis regarding the clinical outcomes of COVID-19 in patients with autoimmune disorders, glucocorticoids increased the rates of hospitalization and mortality [10]. Similarly, a report from the COVID-19 Global Rheumatology Alliance identified that a prednisone dose ≥10 mg/day was associated with higher odds of hospitalization (OR, 2.05; 95% CI 1.06–3.96) [16]. In our study, a threshold of 15 mg/day of oral corticosteroids had the best predictive capacity for SARS-CoV-2 infection occurrence with a sensitivity of 60% (95%CI, 26.2–87.8%), specificity of 91.6% (95%, 80–97.7%), LR+ of 7.2 (95%CI, 2.48–20.91) and LR- of 0.44 (95%, 0.2–0.94). Subsequently, after multivariate adjustment, a current oral corticosteroid dose ≥ 15 mg/day was independently associated with a 7.69-fold higher risk for SARS-CoV-2 infection (OR, 7.69; 95%, 1.3–45.46). Second, the use of hydroxychloroquine was significantly associated with a 91% lower risk for SARS-CoV-2 infection (OR, 0.09; 95%CI, 0.01–0.59). Given that the IFN signaling in LN nephritis closely resembles the immune response to viral infections, the use of hydroxychloroquine, an immunomodulator drug that has been reported to possess antiviral activity in vitro, may lower the risk for severe SARS-CoV-2 infection in patients with SLE (with or without LN) [20]. On the contrary, hydroxychloroquine use either as pre-exposure prophylaxis or as a treatment did not impact COVID-19 outcomes in the general population [21,22,23]. Similarly, reports from patients with SLE showed that hydroxychloroquine treatment did not prevent COVID-19 [8,16,20]. Nonetheless, our data are more hypothesis-generating than definitive proof of the protective effect of hydroxychloroquine for SARS-CoV-2 infection in LN, and the association should not be interpreted as a proof of causality. However, the low incidence and the overall mild forms of COVID-19 might be explained by the high percentage of patients on hydroxychloroquine treatment. A similar hypothesis may be generated from the trend towards an increasing use of the multitarget regimen in LN (that includes calcineurin inhibitors) [24]. Calcineurin inhibitors, especially cyclosporine, have been shown to inhibit the viral replication in vitro (including of several coronaviruses) and may exhibit a protective effect towards SARS-CoV-2 infection [25,26]. Nonetheless, in our study, approximately 10% of patients were on calcineurin inhibitors, a percentage too small to substantiate this hypothesis.

Our study has several limitations that need to be acknowledged. First of all, this is a single-center study, and these findings need to be validated in other populations of patients with LN. Second, we cannot exclude with complete certainty that patients lost to follow-up did not have a SARS-CoV2 infection or even an infection-related death. However, as compared to other studies, this is an actively monitored cohort with periodic clinical and laboratory assessments that also included viral testing through RT-PCR or antigen tests. In addition, our study had a longer period of observation compared to previously reported cohorts. In addition, although the study reflects the characteristics of SARS-CoV-2 infection in a mainly unvaccinated cohort, these results are still important in the context of the dynamics of the pandemic with the emergence of new variants and, moreover, in terms of the future possibility of other outspreads of new viruses to which these patients have not been exposed to.

## 5. Conclusions

In summary, SARS-CoV-2 infection-associated morbidity might be only moderately increased in patients with LN. The current oral corticosteroid dose was the only independent predictor of infection occurrence, while the use of hydroxychloroquine was associated with a protective effect.

## Figures and Tables

**Figure 1 biomedicines-10-02423-f001:**
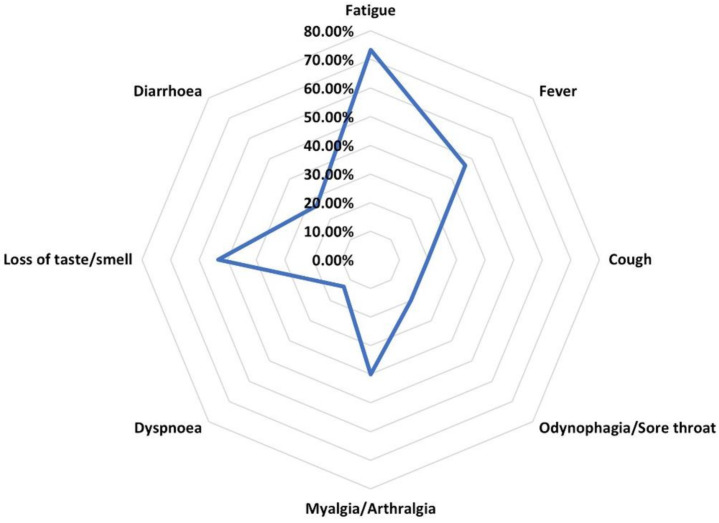
Prevalence of COVID-19 symptoms in patients with LN.

**Figure 2 biomedicines-10-02423-f002:**
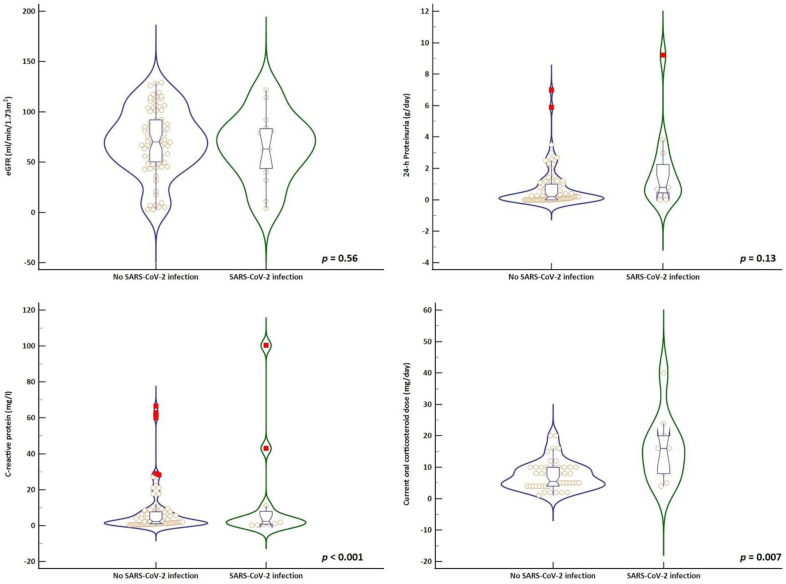
Clinical and laboratory variables in relation with SARS-CoV-2 infection.

**Figure 3 biomedicines-10-02423-f003:**
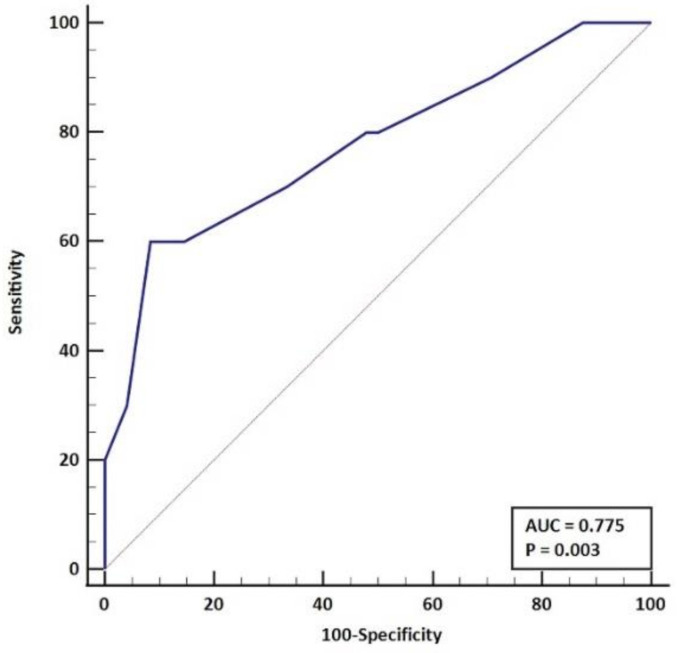
Sensitivity analysis to identify the corticosteroid dose associated with infection occurrence.

**Figure 4 biomedicines-10-02423-f004:**
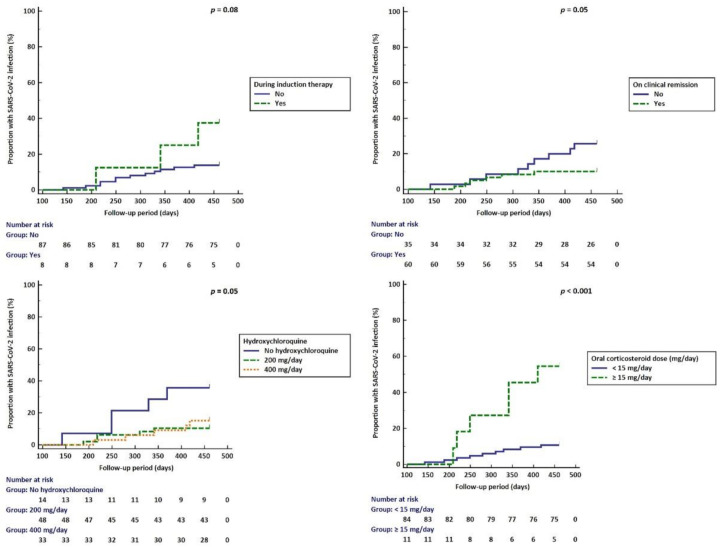
Kaplan–Meier curves showing the impact of timing and type immunosuppressive therapy on SARS-CoV-2 occurrence.

**Table 1 biomedicines-10-02423-t001:** SARS-CoV-2 infection characteristics.

Variable	Value
Number of patients	95
Number of patients with COVID-19 (n, %)	15 (15.8%)
**Symptoms**	
Fatigue (%)	73.3%
Fever (%)	46.7%
Cough (%)	20%
Odynophagia/sore throat (%)	20%
Myalgia/arthralgia (%)	40%
Dyspnea (%)	13.3%
Loss of taste/smell (%)	53.3%
Diarrhea (%)	26.7%
**Outcome**	
Hospitalization (%)	40%
Length of hospitalization (days)	11.5 (IQR, 3.75–14)
Non-invasive ventilation (%)	0%
Invasive mechanical ventilation (%)	0%
Death due to COVID-19 (%)	0%
**Severity of SARS-CoV-2 infection**	
Mild (%)	73.3%
Moderate (%)	26.7%
Severe (%)	0%
**LN evolution following SARS-CoV-2 infection**	
Increase in SCr of at least 0.5 mg/dL (%)	20%
Increase in proteinuria (%)	13.3%
Increase in hematuria (%)	6.6%

Abbreviations: LN, lupus nephritis; SCr, serum creatinine; IQR, interquartile range.

**Table 2 biomedicines-10-02423-t002:** Baseline characteristics of study cohort and factors associated with SARS-CoV-2 infection.

Variable	Entire Cohort	SARS-CoV-2	No SARS-CoV-2	*p*-Value
Number of patients	95	15	80	
Sex (% females)	83.2%	86.7%	82.3%	0.83
Age (years)	39 ± 14	38 ± 12	40 ± 14	0.77
Arterial hypertension (%)	45.3%	53.3%	43.8%	0.49
Diabetes (%)	9.5%	0%	11.2%	0.34
Duration of SLE (years)	7.3 (IQR: 2.3–13.6)	6.7 (IQR: 2.6–10.7)	7.3 (IQR: 2.3–15.3)	0.25
**Extrarenal involvement**				
Cardiac	3.2%	0%	3.8%	0.99
Neurological	11.6%	13.3%	11.2%	0.68
Hematological	65%	73.3%	65%	0.53
Articular	67.4%	46.7%	71.2%	0.07
Serositis	22.1%	20%	22.5%	0.99
Cutaneous	61.1%	40%	65%	0.06
Gastro-intestinal tract	8.4%	6.7%	8.8%	0.99
**Laboratory data**				
Serum creatinine (mg/dL)	1.6 ± 1.9	1.87 ± 2.26	1.63 ± 1.84	0.97
eGFR (mL/min)	69 ± 32	64 ± 33	70 ± 32	0.56
Serum albumin (g/dL)	4 ± 0.85	3.5 ± 1.2	4.1 ± 0.76	0.09
Fibrinogen (mg/dL)	418 ± 121	430 ± 169	415 ± 112	0.75
Hematuria (cells/mmc)	10 (IQR: 4.3–30.2)	29 (IQR: 10–118)	10 (IQR: 4–26.7)	0.01
24-h proteinuria (g/day)	0.3 (IQR: 0–1.2)	0.8 (IQR: 0.4–2.5)	0.2 (IQR: 0–1)	0.13
Leukocytes (cells/mmc)	7411 ± 3174	8702 ± 2465	7169 ± 3246	0.04
Neutrophils (cell/mmc)	5007 ± 2447	6282 ± 2377	4768 ± 2399	0.03
Lymphocytes (cells/mmc)	1647 ± 954	1593 ± 789	1657 ± 986	0.78
Hemoglobin (g/dL)	12.1 ± 1.9	12.4 ± 1.9	12.1 ± 1.9	0.52
Serum IgA (mg/dL)	250 (IQR: 153–337)	224 (IQR: 184–392)	250 (IQR: 141–333)	0.54
Serum IgG (mg/dL)	1141 (IQR: 811–1519)	1260 (IQR: 573–1550)	1130 (IQR: 824–1519)	0.88
Serum IgM (mg/dL)	86 (IQR: 47–157)	63 (IQR: 37–149)	89 (IQR: 47–166)	0.6
C-reactive protein (mg/L)	8.9 ± 16.8	14.3 ± 29.5	8 ± 13.7	<0.001
Serum C3 (mg/dL)	98 ± 30	96 ± 31	98 ± 30	0.81
Serum C4 (mg/dL)	19 ± 9	18 ± 7	19 ± 9	0.63
**Immunosuppression**				
On immunosuppression (%)	83.2%	80%	83.8%	0.71
During induction (%)	8.4%	20%	6.2%	0.11
During maintenance (%)	74.7%	60%	77.5%	0.19
Hydroxychloroquine (%)	85.3%	66.7%	88.8%	0.04
Hydroxychloroquine dose (%)				
• 200 mg	50.5%	33.3%	53.8%	0.07
• 400 mg	34.7%	33.3%	35%	
Calcineurin inhibitor (%)	9.5%	13.3%	8.8%	0.63
Mycophenolate mofetil (%)	34.7%	46.7%	32.5%	0.29
Mycophenolate mofetil dose (g/day)	1 (IQR: 1–1.5)	1 (IQR: 1–1.5)	1 (IQR: 1–1.5)	0.67
Cumulative cyclophosphamide dose (g)	3 (IQR: 0.125–7.5)	2.87 (IQR: 1.1–7.1)	3 (IQR: 0–8.75)	0.75
Rituximab (%)	10.5%	13.3%	10%	0.65
Azathioprine (%)	15.8%	6.7%	17.6%	0.45
Oral corticosteroids (%)	60%	66.7%	58.8%	0.56
Corticosteroid dose (mg/day)	8 (IQR: 5–10)	16 (IQR: 7–21)	6 (IQR: 4–10)	0.007
On clinical remission (%)	63.2%	40%	67.5%	0.04

Abbreviations: SLE, systemic lupus erythematosus; eGFR, estimated glomerular filtration rate; IQR, interquartile range.

**Table 3 biomedicines-10-02423-t003:** Logistic regression analysis regarding risk factors associated with SARS-CoV-2 infection.

Variable	Univariate Analysis		Multivariate Analysis	
	Odds Ratio (95%CI)	*p* Value	Odds Ratio (95%CI)	*p* Value
eGFR (for 1 mL/min/1.73m^2^)	0.99 (0.97–1.012)	0.54	-	-
Serum albumin (for 1 g/dL)	0.55 (0.31–0.96)	0.03	1.04 (0.35–3.03)	0.94
C-reactive protein (for 1 mg/L)	1.01 (0.98–1.04)	0.25	1.006 (0.97–1.04)	0.75
Hematuria (for 1 cell/mmc)	1.00 (0.99–1.001)	0.8	-	-
On immunosuppression (yes vs. no)	0.77 (0.19–3.14)	0.72	-	-
During induction therapy (yes vs. no)	3.75 (0.79–17.7)	0.09	3.91 (0.56–30.4)	0.1
During maintenance therapy (yes vs. no)	0.43 (0.13–1.38)	0.16	-	-
On hydroxychloroquine (yes vs. no)	0.25 (0.07–0.91)	0.03	0.09 (0.01–0.59)	0.01
Hydroxychloroquine dose (vs. no HCQ)	-	-	-	-
• 200 mg	0.2 (0.05–0.87)	0.03	-	-
• 400 mg	0.32 (0.07–1.36)	0.12	-	-
On CNI (yes vs. no)	1.6 (0.29–8.59)	0.58	-	-
On MMF (yes vs. no)	1.81 (0.59–5.55)	0.29	-	-
On azathioprine (yes vs. no)	0.33 (0.04–2.77)	0.31	-	-
On rituximab (yes vs. no)	1.38 (0.26–7.26)	0.7	-	-
Current oral corticosteroid dose (≥15 mg/d vs. <15 mg/d)	10 (2.53–39.49)	0.005	7.69 (1.3–45.46)	0.02
Cumulative cyclophosphamide dose (≥3 g vs. <3 g)	0.75 (0.22–2.6)	0.65	-	-
With clinical remission (yes vs. no)	0.27 (0.08–0.85)	0.02	0.28 (0.05–1.45)	0.72

Abbreviations: eGFR, estimated glomerular filtration rate; HCQ, hydroxychloroquine; CNI, calcineurin inhibitor; MMF, mycophenolate mofetil.

## Data Availability

The data presented in this study are available in “The characteristics of SARS-CoV-2 infection in an actively monitored cohort of patients with lupus nephritis”.

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
