# Peer review of "Characteristics of SARS-CoV-2 Infection in an Actively Monitored Cohort of Patients with Lupus Nephritis"

_biomedicines, 2022, doi:10.3390/biomedicines10102423_

Round 1

Reviewer 1 Report

Thank you for giving me the opportunity to read and comment a report “Characteristics of SARS-CoV-2 infection in an actively monitored cohort of patients with lupus nephritis.”, by Obrișcă B, et al.

In the reviewed manuscript, the characteristics of SARS-CoV-2 infection and the associated risk factors in an actively monitored cohort of patients with LN throughout an observation period spanned over the first three waves of COVID-19 pandemic has been investigated.

This paper is well written, correctly structured with a suitable research concept, the study limitations are addressed, and it is of relevance to readers of the journal. However, I include a few comments for your consideration.

·       According to the rules of the journal the abstract must be without headings.

·       It would be desirable to add the meaning of the abbreviation IQR in tables 1 and 2.

·       In the opinion of this reviewer, a more in-depth discussion of the protective effect of hydroxychloroquine is needed since it is not currently part of the pharmacological treatment strategies for COVID-19.

·        The study includes data from February 2020 to May 2020, however, there is now over 1 year of data that was not included. We know that pandemic evolution has changed greatly during this time, especially with the occurrence of new virus variants, therefore I am unclear how relevant the results of this study are in the current time frame. I would like that the authors to discuss if the results obtained in the first three waves of the COVID-19 outbreak could be extrapolated to present.

Author Response

Dear Biomedicines Editorial team,

On behalf of the co-authors I want to thank you for the opportunity of incorporating reviewer comments made in relation to our manuscript entitled “Characteristics of SARS-CoV-2 infection in an actively monitored cohort of patients with lupus nephritis”. We hope to have addressed all the comments and suggestions and believe that it has made our report more clear and meaningful for publication.

Reviewer 1

Thank you for giving me the opportunity to read and comment a report “Characteristics of SARS-CoV-2 infection in an actively monitored cohort of patients with lupus nephritis.”, by Obrișcă B, et al.

In the reviewed manuscript, the characteristics of SARS-CoV-2 infection and the associated risk factors in an actively monitored cohort of patients with LN throughout an observation period spanned over the first three waves of COVID-19 pandemic has been investigated.

This paper is well written, correctly structured with a suitable research concept, the study limitations are addressed, and it is of relevance to readers of the journal. However, I include a few comments for your consideration.

R: Thank you for the appreciation and constructive comments.

  • According to the rules of the journal the abstract must be without headings.

R: We have re-adjusted the abstract.

  • It would be desirable to add the meaning of the abbreviation IQR in tables 1 and 2.

R: Thank you for the observation. We have added the abbreviation in the Table.

  • In the opinion of this reviewer, a more in-depth discussion of the protective effect of hydroxychloroquine is needed since it is not currently part of the pharmacological treatment strategies for COVID-19.

R: Our results are rather hypothesis generating than definitive proof. We have stated that use of HCQ might posses and antiviral activity in vitro. However, given that there is only weak evidence for this aspect, we have further stressed that this is only an association and not proof of causality in order not to overstate our results.

  • The study includes data from February 2020 to May 2020, however, there is now over 1 year of data that was not included. We know that pandemic evolution has changed greatly during this time, especially with the occurrence of new virus variants, therefore I am unclear how relevant the results of this study are in the current time frame. I would like that the authors to discuss if the results obtained in the first three waves of the COVID-19 outbreak could be extrapolated to present.

R: Thank you for the observation. Actually, the follow-up date is up to May 2021. Although we agree that the pandemic evolution has greatly changed during the last year, it is becoming increasingly acknowledged that the previous vaccination confers less protection to newer SARS-CoV-2 variants and given that this study describes the SARS-CoV-2 infection in a mainly unvaccinated cohort (we did not include in the analysis the vaccination status as it was an ongoing process; actually 41.5% of study cohort was vaccinated, and patients that developed a SARS-CoV-2 infection more likely received a vaccine after the infection development compared to patients that did not develop an infection. Given this aspect, we identified a potential misleading bias if this aspect would have been introduced in the multivariate analysis). For the sake of clarity we have updated this part. Given that this cohort was mainly an unvaccinated cohort, these results are still relevant for future SARS-CoV-2 variants or, as well, other newly emerging viral infections, for which a particular cohort of patients with an autoimmune disease undergoing immunosuppressive therapy has not been exposed to. We have updated the discussion according to your suggestion.

We hope that we have addressed all the issues of your comments.

Sincerely Yours,

Bogdan Obrisca MD, PhD

Corresponding author: Bogdan Obrisca – Department of Nephrology, Fundeni Clinical Institute,

258 Fundeni Street, District 2, Bucharest, Romania, zip code 022328; [email protected]

Reviewer 2 Report

The manuscript submitted by Bogdan Obrișcă describes a characteristic of SARS-CoV-2 infection in an actively monitored cohort of patients with lupus nephritis. This work is well written and very clear.

Author Response

Dear Biomedicines Editorial team,

On behalf of the co-authors I want to thank you for the opportunity of incorporating reviewer comments made in relation to our manuscript entitled “Characteristics of SARS-CoV-2 infection in an actively     monitored cohort of patients with lupus nephritis”. We hope to have addressed all the comments and suggestions and believe that it has made our report more clear and meaningful for publication.

Reviewer 2

The manuscript submitted by Bogdan Obrișcă describes a characteristic of SARS-CoV-2 infection in an actively monitored cohort of patients with lupus nephritis. This work is well written and very clear.

R: Thank you for the appreciation!

We hope that we have addressed all the issues of your comments.

Sincerely Yours,

Bogdan Obrisca MD, PhD

Corresponding author: Bogdan Obrisca – Department of Nephrology, Fundeni Clinical Institute,

258 Fundeni Street, District 2, Bucharest, Romania, zip code 022328; [email protected]